# Diagnostic Challenge of Invasive Lobular Carcinoma of the Breast: What Is the News? Breast Magnetic Resonance Imaging and Emerging Role of Contrast-Enhanced Spectral Mammography

**DOI:** 10.3390/jpm12060867

**Published:** 2022-05-25

**Authors:** Melania Costantini, Rino Aldo Montella, Maria Paola Fadda, Vincenzo Tondolo, Gianluca Franceschini, Sonia Bove, Giorgia Garganese, Pierluigi Maria Rinaldi

**Affiliations:** 1Radiology Unit, Mater Olbia Hospital (Qatar Foundation Endowment and Policlinico Universitario Agostino Gemelli IRCCS Foundation), Strada Statale 125 Orientale Sarda, 07026 Olbia, Italy; melania.costantini@materolbia.com (M.C.); mariapaola.fadda@materolbia.com (M.P.F.); pierluigi.rinaldi@materolbia.com (P.M.R.); 2Dipartimento Diagnostica per Immagini, Radioterapia Oncologica ed Ematologia, Fondazione Policlinico Universitario Agostino Gemelli IRCCS, Area Diagnostica per Immagini, 00168 Rome, Italy; 3General Surgery Unit, Mater Olbia Hospital (Qatar Foundation Endowment and Policlinico Universitario Agostino Gemelli IRCCS Foundation), Strada Statale 125 Orientale Sarda, 07026 Olbia, Italy; vincenzo.tondolo@materolbia.com; 4Multidisciplinary Breast Center, Dipartimento Scienze Della Salute Della Donna e del Bambino e di Sanità Pubblica, Fondazione Policlinico Universitario Agostino Gemelli IRCCS, 00168 Rome, Italy; gianluca.franceschini@policlinicogemelli.it; 5Istituto di Semeiotica Chirurgica, Università Cattolica del Sacro Cuore, 00168 Rome, Italy; 6Gynecology and Breast Care Center, Mater Olbia Hospital (Qatar Foundation Endowment and Policlinico Universitario Agostino Gemelli IRCCS Foundation), Strada Statale 125 Orientale Sarda, 07026 Olbia, Italy; sonia.bove@materolbia.com (S.B.); giorgia.garganese@materolbia.com (G.G.); 7Dipartimento Scienze Della Vita e Sanità Pubblica, Sezione Ginecologia e Ostetricia, Università Cattolica del Sacro Cuore, 00168 Rome, Italy

**Keywords:** breast cancer, invasive lobular carcinoma, breast magnetic resonance imaging, contrast-enhanced spectral mammography

## Abstract

Invasive lobular carcinoma is the second most common histologic form of breast cancer, representing 5% to 15% of all invasive breast cancers. Due to an insidious proliferative pattern, invasive lobular carcinoma remains clinically and radiologically elusive in many cases. Breast magnetic resonance imaging (MR) is considered the most accurate imaging modality in detecting and staging invasive lobular carcinoma and it is strongly recommended in pre-operative planning for all ILC. Contrast-enhanced spectral mammography (CESM) is a new diagnostic method that enables the accurate detection of malignant breast lesions similar to that of breast MR. CESM is also a promising breast imaging method for planning surgeries. In this study, we compare the ability of contrast-enhanced spectral mammography (CESM) with breast MR in the preoperative assessment of the extent of invasive lobular carcinoma. All patients with proven invasive lobular carcinoma treated in our breast cancer center underwent preoperative breast MRI and CESM. Images were reviewed by two dedicated breast radiologists and results were compared to the reference standard histopathology. CESM was similar and in some cases more accurate than breast MR in assessing the extent of disease in invasive lobular cancers. Further evaluation in larger prospective randomized trials is needed to validate our preliminary results.

## 1. Introduction

Breast cancer (BC) affects millions of women worldwide. Invasive lobular carcinoma (ILC) is the second most common type of invasive breast cancer accounting for 5–15% of cases [1,2,3]. The incidence of ILC is highest in white older women [4] and over the last two decades it has increased, also due to the use of hormone replacement therapies [5,6,7,8,9].

Generally, diagnosis is late at a more advanced tumoral stage than other BC types [10,11,12,13,14]. In fact, ILC tends to be clinically silent and difficult to detect through imaging due to its underhand growth pattern [15].

ILC is well recognized by pathologists and is composed of relatively small, discohesive epithelial cells infiltrating the fibrous stroma in a single-file pattern with minimal stromal reaction. Pathological hallmark features of ILC include lack or loss of cell–cell adhesion molecule E-cadherin (encoded by CDH1 gene) [2,16,17] and positive for both the oestrogen (ER) and progesterone receptors (PgR) and negative for the human epidermal growth factor receptor 2 (HER-2) [2].

Invasive ductal carcinoma (IDC) is generally identified as a mass on clinical and radiological examinations and as it originates from mammary ducts, microcalcifications are also a common sign on mammography [18]. ILC originates in the lobule structures that show individual layers of cells traversing the surrounding tissues similar to the threads of a spider’s web. This infiltrative growth pattern generally does not induce conspicuous desmoplastic reaction and architectural distortion neither forms a mass. The lesion often shows a mammographic density less than or equal to surrounding breast parenchyma, indistinguishable from normal breast tissue [19] and microcalcifications are a rare mammographic finding (variating widely from 0 to 24%) [20].

These widespread growth patterns make diagnosis particularly challenging.

Digital mammography is the first level imaging examination for breast cancer diagnosis with a sensitivity between 63 and 98% [1]. However, its diagnostic performance in ILC detection is more limited ranging between 57–81% [9], especially in dense breast tissue (in extremely dense breast tissue, ILC detection can be as low as 30%) [17].

Digital breast tomosynthesis (DBT) has the potential to explore breast tissues by producing thin slices of the mammographic view. DBT reduces the tissue-masking effect and improves lesion conspicuity with a better evaluation of parenchymal distortion, asymmetries and ill-defined masses, which are common findings in ILC [21,22]. Despite its advantages, DBT is based on differential tissue X-ray attenuation and can remain suboptimal.

Ultrasound (US) is not used as a screening tool and generally evaluates suspicious findings detected at clinic or at mammographic examinations.

All these imaging diagnostic techniques can help radiologists detect equivocal malignant signs but often a correct diagnosis can be challenging even at later stages when tumors are larger, multifocal, multicentric and metastatic.

Therefore, it is important to improve breast imaging methods to detect and diagnose ILC.

At present, breast magnetic resonance imaging (BMR) is recognized as the most sensitive diagnostic tool to detect and stage ILC. BMR provides information about the morphology of the lesion but also an analysis of tumor neoangiogenesis. BMR is quite useful in patients in whom the diagnosis of ILC is proved and the disease extent is uncertain from physical exam and mammography/ultrasound tests. BMR may detect additional diseases not otherwise appreciated through conventional imaging and may provide more accurate staging information to guide surgical treatment [23,24].

Recently a new emerging digital mammography technology based on contrast enhancement evaluation, contrast-enhanced spectral mammography (CESM), is improving cancer detection and decreasing misdiagnosis rates [25].

CESM combines conventional mammography with the intravenous administration of an iodinated contrast material offering both morphological and functional information of breast tissue [26,27,28].

Several studies have shown that the diagnostic performance of CESM is similar to that of MRI and that CESM may be useful for indications previously reserved for MRI [25,29].

In our breast diagnostic unit, CESM has been used since May 2019 and our experience is showing the new potential in helping diagnosis in different clinical settings.

In this study, we compare the ability of CESM and BMR in the detection and staging of invasive lobular carcinoma to investigate the potential of CESM in the diagnostic work-up of a patient with ILC.

## 2. Materials and Methods

### 2.1. Study Population

This is a single-institution prospective clinical study approved by the Institutional Review Board of Mater Olbia Hospital.

All participants received diagnostic procedures as part of their routine medical care and provided written informed consent before examinations.

The study duration was 24 months starting from September 2019 to September 2021.

In this period, all patients with a biopsy-proven invasive lobular carcinoma of the breast were enrolled in the study.

Eligibility criteria were:Patients who could undergo CESM and BMR within 1 month in our breast radiological section;Patients treated with prior surgery at our hospital;Patients whose final diagnosis was obtained in our pathology unit.

Exclusion criteria were:Patients who were pregnant or lactating;Patients with general contraindication to CESM or MR.

Both CESM and BMR were performed in accordance with the relevant guidelines and regulations [24,30,31].

Informed consent was obtained for both procedures.

### 2.2. Contrast-Enhanced Spectral Mammography (CESM)

All CESM exams were performed with a Senographe Pristina Mammography System (GE Healthcare, Chicago, IL, USA) integrated with SenoBright HD equipment (GE Healthcare) for dual-energy CESM acquisition.

An experienced mammography technologist prepared the patient by explaining the times of the procedure. A nurse prepared the patient for contrast medium intravenous injection placing a catheter into the antecubital vein of the arm contralateral to the affected breast.

Non-ionic contrast agent Iohexol (Omnipaque 350, GE Healthcare) at a dose of 1.5 mL/Kg and at a flow of 3 mL/s followed by a bolus of 30 mL of saline solution was used.

Approximately 2 min after contrast agent injection, standard mammographic views were obtained in the following order: the craniocaudal (CC) projection of the unaffected breast, the CC projection of the affected breast, the mediolateral oblique (MLO) projection of the affected breast and finally the MLO projections of the unaffected breast. The radiographic process was completed in 7–8 min.

The mammography unit in the CESM mode automatically performed a low-energy and a high-energy exposure in each view. A processing software generated subtracted images in each view immediately showing contrast agent uptake information.

Both low-energy and subtracted images in each view were transferred to the workstation for radiologist evaluation. The Seno Iris mammography workstation (GE Healthcare) was used for viewing images.

### 2.3. Breast Magnetic Resonance Imaging (BMR)

BMR examinations were performed with 1.5 T MR scanner system (Signa HD, GE Healthcare) using a dedicated 8-channel breast coil.

Our protocol included the following sequences:Three-plane localizer images;Axial T2-weighted sequence;Axial diffusion weighted echo planar imaging with sensitizing diffusion gradients applied sequentially in the x-, y- and z-directions with *b* values of 0 and 1000 s/mm^2^;T1-weighted fat saturation dynamic axial sequence (Vibrant) including one pre-contrast acquisition and five post-contrast. Post-contrast dynamic MR images were acquired after the administration of 0.2 mL/kg gadolinium contrast agent (Gadoteric acid, Claricyclic, GE Healthcare) a flow of 2 mL/s followed by a bolus of 20 mL of saline solution;T1-weighted fat saturation sagittal sequence.

In premenopausal women, BMR was preferably performed on day 6–13 of the menstrual cycle, even when oral contraception is used [32].

### 2.4. Pathology

Histopathological examination was performed in our histopathology laboratory by 2 pathologists experienced in breast cancer diagnostics.

All specimens were fixed, embedded and sectioned. Conventional hematoxylin-eosin staining and immunohistochemical analysis was carried out according to the World Health Organization Pathological Classification and Diagnostic Criteria of Breast Tumors (2019) [33].

Classic and pleomorphic ILC represent subtypes of ILC, classified according to the histological grow modalities and cytomorphological features [33].

Loss of E-cadherin expression is typical of ILC, in contrast to DCI, a specifical characteristic that is important in the diagnosis and classification of ILC. In fact, about 90% of ILC tumors lack this cell–cell adhesion molecule (CAM), essential for cell viability. Lack of E-cadherin causes the dysregulation of the molecule, giving the typical discohesive growth aspect of ILC [33,34]. Additionally, in pleomorphic lobular carcinomas, a complete loss of E-cadherin is observed [34].

Dysregulation of CDH1 gene (chromosome 16q22) has been identified as responsible for the loss of E-cadherin.

Differential diagnosis between ductal and lobular cancers can be evaluated with the immunohistochemical detection of E-cadherin, even in difficult cases with similar or equivocal histological characteristics [33].

Histologically, typical features of ILC include small monomorphic cells, lack of cell–cell cohesion and round or notched ovoid nuclei surrounded by a tiny layer of cytoplasm [34]. ER and PgR status and HER2 over-expression were determined through immunohistochemical staining according to the ASCO/CAP guidelines [35,36].

ILC is often associated with good prognosis and low histological grade, typically being strongly ER positive, HER-2 negative and showing a low rate of ERBB2 amplification. Although ERBB2 amplifications and mutations can be observed in 8% of ILCs, these are more frequent in pleomorphic forms, depicting a poorer prognosis [34]. Ki-67 values were measured as the percentage of malignant cells stained positively with the antihuman Ki-67 monoclonal antibody MIB1 according to current recommendations [37].

### 2.5. Data Analysis

Pathological stage (pTNM) and biomarker status (ER, PgR, HER-2, Ki-67, tumor grade) of cancer were collected from the pathology database of our hospital.

CESM and BMR images were evaluated in consensus by two dedicated radiologists with more than 15 years of experience in breast imaging.

Breast Imaging Reporting and Data System (BI-RADS) lexicon was used to assess BMR images [38].

CESM was assessed using a BI-RADS-like classification according to current recommendations [31].

Results of CESM examination includes the results of the conventional mammographic images (using the Breast Imaging Reporting and Data System mammography lexicon) and the iodine-enhanced images (using part of the Breast Imaging Reporting and Data System MRI lexicon) [38].

Presence of pathological enhancement, type of enhancement (mass or non-mass), number of enhanced lesions and size of the greater lesions (namely target lesion) were assessed for each imaging modality.

Multifocal and multicentric breast cancers are defined as the presence of two or more tumors within the same breast. In radiological assessment, if the distance between the lesions is lower than/equal 5 cm, the cancer is defined as multifocal, and when the distance is higher than 5 cm, the cancer is defined as multicentral [39,40]. For the present study, multifocal and multicentric cancers were commonly defined as multifocal–multicentric cancers (MFMCC).

Imaging data were compared to pathological results.

For statistical analysis of the pathology report, we considered the maximum diameter of target lesion and the presence of MFMC. The sensitivity of CESM and BMR in detecting target lesions (previously identified through conventional breast imaging examination and histologically proved as CLI) is defined as the proportion of cases in which a target lesion is correctly visualized as pathological enhancement area.

The sensitivity of CESM and BMR in correctly staging CLI is defined as the proportion of cases of proved MFMC disease (as described in definitive pathological report) in which more areas of pathological enhancement were identified.

Statistical analysis and graphic visualization were obtained with the use of SPSS-Pasw Statistics (SPSS, Chicago, IL, USA).

## 3. Results

In the study period, 257 new cases of breast cancer were treated in our hospital.

Breast cancer lesions were detected during mammography or ultrasound examination and US-guided core needle biopsy or mammographic-guided VABB (vacuum assisted breast biopsy) was performed for initial histopathological diagnosis. Biopsy results showed 41 cases of invasive lobular carcinoma accounting for 15.9% of total breast cancers.

One patient was excluded because of MR general contraindication (cardiac implantable electronic device).

Two patients were excluded because neoadjuvant chemotherapy was the prior indicated treatment.

In total, 38 consecutive CLI patients were definitively enrolled for this study.

All patients received both CESM and BMR examinations prior to have surgery at our hospital.

Patient median age was 59.7 years (range 40–85 years).

As decided in the Breast Multidisciplinary Tumor Board, 25 patients had breast conserving surgery and 13 patients had mastectomy.

The final pathological results confirmed in all cases an invasive lobular carcinoma (including 10 cases of pleomorphic forms) and revealed 16 cases of multifocal/multicentric diseases. There was also a case of bilateral disease in which contralateral breast cancer was an invasive ductal carcinoma.

Mean size of the target lesion was 23.78 mm (range: 6–75 mm).

Most of the lesions were G2. No cases of G1 lesions were documented. The only five cases of G3 lesions were pleomorphic variants (two cases of triple negative and three cases of luminal B).

Stage of disease was T1 in 20 cases, T2 in 16 cases and T3 in 2 cases.

Concerning nodal status, there were 27 N0, 8 N1 and 3 N2 cases.

Molecular characteristics according to St. Gallen Classification (Gnant M, 2015) showed 22 cases of luminal A, 14 cases of luminal B and 2 cases of triple-negative breast cancers.

Patient characteristics are reported in Table 1.

No adverse reaction was observed either to iodinated contrast medium or to gadolinium contrast agent.

Results concerning sensitivity in CESM and BMR in detecting both target lesions and MFMC disease are shown in Table 2.

BMR detects all target lesions demonstrating 100% sensitivity.

CESM only missed one target lesion demonstrating 97.36% sensitivity. In this case, no conspicuous enhancement is shown at the time of CESM evaluation, but a small distortion was identified prior at tomosynthesis projection during the screening mammographic exam.

BMR missed two cases of MFMC disease (Figure 1). Sensitivity of BMR in correctly staging multifocal/multicentric disease was 94.73%. In both these cases, the target lesion was mass-like.

CESM missed one case of multifocal disease. Sensitivity of CESM in correctly staging multifocal/multicentric disease was 94.73%.

Both CESM and BMR correctly identify a contralateral breast cancer (an invasive ductal carcinoma, no specific type) (Figure 2).

BMR showed a case of false positive contralateral enhancement.

The means of the maximum diameter of target lesions assessed by BMR and CESM compared to histopathology are reported in Table 3.

The Pearson correlation coefficients of BMR and CESM versus micro-histology were statistically significant (BMR vs. Micro-histology, r = 0.945, *p* < 0.001; CESM vs. Micro-histology, r = 0.937, *p* < 0.001) (Figure 3).

Lesion sizes determined with CESM and breast MRI were similar (Figure 2), but were larger than those from histopathological examination.

BMR showed 25 cases of pathological mass enhancement and 13 cases of non-mass enhancement. CESM showed 24 cases of mass enhancement and 13 cases of non-mass enhancement (Figure 3 and Figure 4).

Pleomorphic forms often presented as non-mass lesions (8/10 in our case series).

The flowchart diagram reported in Figure 4 clarifies the strategy of this study.

## 4. Discussion

Invasive lobular carcinoma is an independent entity among breast cancers with specific epidemiological, biological and clinical characteristics [1,2,3,4].

Invasive lobular carcinoma can be difficult to detect at clinical examinations and at standard breast imaging examinations because it often exhibits an infiltrative growth pattern not forming a mass. In these cases, it is not uncommon for mammographic and ultrasound examinations to lead to false negative results [10,11,12,13,14,15].

As the performance of conventional breast imaging modalities and clinical breast examinations is limited, most guidelines recommend BMR for staging of invasive lobular cancers [23,24].

MRI is a valuable diagnostic tool in breast imaging and it is recognized as the most accurate imaging test to diagnose and stage ILC [23,24].

It offers a multiparametric approach evaluating both morphologic and enhancement characteristics of a tissue. The use of contrast medium permits us to identify mass and non-mass enhancement alterations improving its diagnostic potential [41,42,43].

Despite this, even BMR can fail the detection of certain tumors that demonstrate little or no enhancement, generally low aggressive forms. Sensitivity of BMR in the detection of ILC is reported to reach 91–100%, which is higher than with US and mammography [41,42,43].

At present another breast imaging method, contrast-enhanced spectral mammography, is able to evaluate contrast uptake alterations. CESM provides morphological information similarly to conventional mammography and detects breast-enhancing lesions similar to breast magnetic resonance imaging (MRI) using an iodinate contrast medium.

Several studies have already shown that the diagnostic performance of CESM in detecting and staging breast cancer appears to be similar to BMR [44,45].

Our study investigates the ability of CESM and BMR in the detection of invasive lobular carcinoma.

BMR and CESM findings of ILC include mass and non-mass enhancement alterations.

The Breast Imaging Reporting and Data System lexicon (ACR MR-BI-RADS Atlas) cites that the non-mass enhancement lesion is an area whose internal enhancement characteristics can be distinguished from the normal surrounding breast parenchyma, without an associated mass [38].

In our study, BMR detects all ILC showing 100% sensitivity in detecting this type of tumor.

CESM only missed one target lesion. There was a small irregular mass correctly detected through BMR and visualized as focal distortion in tomosynthesis projection performed during the screening mammographic exam.

According to literature data [41,42,43], the only two radiological findings of ILC were irregular masses and non-mass enhancement areas.

No cases of circumscribed masses were observed.

Non-mass enhancement cases were more often a clumped or clustered ring type.

We also observed that non-mass lesions frequently correspond to a pleomorphic variant of ILC.

In CESM, the type of enhancement (mass or non-mass) was the same observed on BMR.

However, there are some differences between the two methods.

In CESM, all mass lesions demonstrated non-circumscribed margins and heterogeneous internal enhancement but in some cases the image appears blurrier than in BMR and any further categorization becomes difficult. In these cases, mammographic image can help the analysis of margin findings.

According to MRI-BI-RADS lexicon, the internal enhancement pattern of non-mass lesions includes homogeneous, heterogeneous, clumped and clustered rings [46]. These descriptors are hard to apply in CESM (Figure 5 and Figure 6).

These difficulties in evaluating some specific descriptors seems to make CESM examination less accurate. However, we adapted the lexicon created for MRI and used it for CESM, thus a new specific CESM lexicon should probably be promoted.

Currently available evidence indicates that multifocal and multicentric breast cancers have a similar prognosis to unifocal cancers provided that the tumors are completely excised.

Therefore, it is very important to correctly stage the loco-regional extension of CLI.

In our study, 42% of patients showed MFMC disease at definitive pathological diagnosis.

Both BMR and CESM showed high sensitivity to the detection of MFMC disease.

BMR missed two cases and CESM missed one case of multifocal disease. In all these cases, the target lesion was an irregular-enhanced mass and an ultrasound helped establish the correct diagnosis, identifying multiple lesions in both cases.

Both CESM and BMR correctly identified a contralateral breast cancer (an invasive ductal carcinoma, no specific type).

Both BMR and CESM have similar potential in the measurement of target lesions. Both methods showed a minimal overestimation of the real pathological size, but these data do not influence the treatment.

CESM is more acceptable and less expensive than BMR.

Many existing mammography units can be upgraded with CESM equipment instead of an MRI scanner, which has a limited availability. Exam time and reading time are very low.

Patients with claustrophobia or implantable devices that are not compatible with MRI, or obese patients, can safely perform the exam.

The radiation dose is negligible.

CESM can detect abnormalities visible on the low-energy mammograms such as microcalcifications, architectural distortion or spiculated lesions that cannot have significant enhancement. However, CESM evaluation is limited to the breast. BMR even allows the evaluation of the axilla, chest wall and internal mammary chain.

Considering our observations above, we can conclude that the combination of all the available imaging methods permits the best diagnosis at the price of more costs and time. Evaluating our experience, we can postulate that an excellent mammography with tomosynthesis and CESM associated with an excellent ultrasound examination in the hands of dedicated experts could reach a very high level of diagnostic accuracy whilst avoiding BMR use.

According to recent literature data [47], our study demonstrated that CESM is an excellent alternative to BMR in diagnosing and staging ILC.

This study has limitations. It is a preliminary single-center study with a small number of cases. Results depend on the unique experience of our breast unit and on the specific population. However, our data agree with the observations made by many authors.

## 5. Conclusions

Invasive lobular carcinoma is a distinct type of breast cancer that poses clinical and diagnostic challenges.

A deeper understanding of the biological behavior and tumor microstructure has improved diagnosis and management.

BMR is the more accurate diagnostic tool in the evaluation of CLI.

CESM has an emerging role in breast imaging. The results of our study showed that CESM is similar to BMR in detecting and staging ILC.

Scientific evidence on the accuracy of CESM is largely based on single-center studies with relatively small case studies. Larger, multicenter prospective trials are needed to confirm our preliminary results.

## Figures and Tables

**Figure 1 jpm-12-00867-f001:**
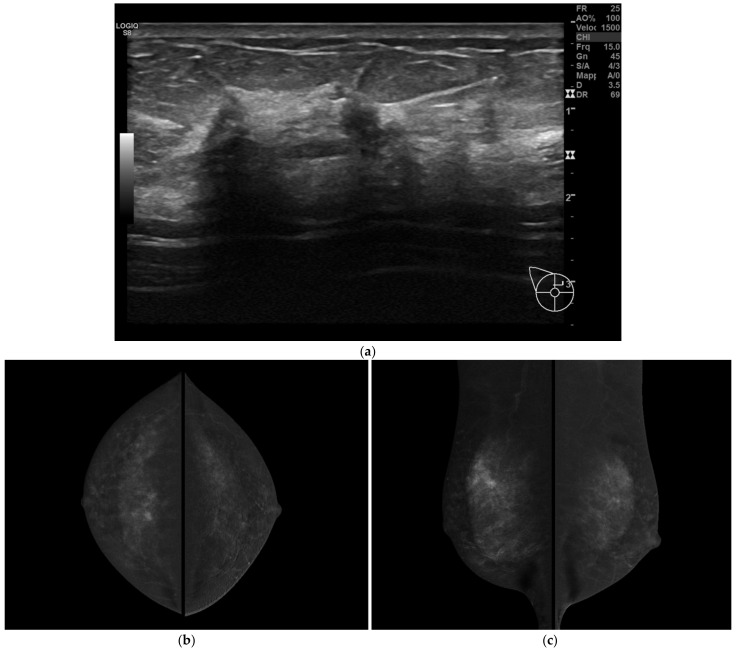
A 47-year-old women with multifocal CLI. Patient performs mammography and ultrasound examinations for annual prevention check. (**a**) At ultrasound examination, two small suspicious hypoechoic lesions are visualized. Needle biopsy was performed on both lesions and invasive lobular carcinoma was diagnosed in both cases. (**b**) CESM CC and (**c**) CESM MLO views show some suspicious small-enhanced masses in the upper quadrants of the right breast suggesting a multifocal disease (T1bm N0 M0, ER 90%, PgR 5%, Ki67 20%, G2). (**d**) BMR axial MIP reconstruction and (**e**) BMR sagittal MIP reconstructions show only one small-enhanced mass lesion. In this case, BMR missed the real extent of the disease. (**f**,**g**) Postoperative specimen histology including one of the lesions previously described, Hematoxilin & Eosin stain. (**f**) Low power magnification showing invasive breast carcinoma with sparse, poorly cohesive neoplastic cells, associated with stromal desmoplasia (magnification 100×). (**g**) At higher magnification, the neoplastic cells are small sized, with scant cytoplasm and mild nuclear pleomorphism, arranged in single cells or small cords, consistent with the diagnosis of CLI (magnification 400×).

**Figure 2 jpm-12-00867-f002:**
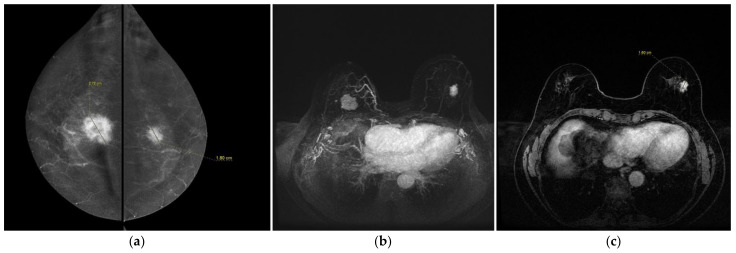
A 74-year-old women with synchronous bilateral breast cancer. Patient performs mammography and ultrasound examinations because a palpable mass in the upper quadrant of the right breast. (**a**) CESM CC views show an irregular-enhanced mass in the right breast and an irregular-enhanced mass in the left breast. They were an invasive ductal carcinoma in the right breast (27 mm in size) and an invasive lobular carcinoma in the left (18 mm in size). (**b**) BMR, axial MIP (Maximum intensity projection) reconstruction shows bilateral breast cancer. Note the similarities with the CESM images (morphology of the masses and increased local vascularity in the right breast). (**c**) BMR, first post-contrast axial T1-weighted fat saturation dynamic axial sequence (Vibrant). Measurements of the lesions show good agreement with CESM (16 mm).

**Figure 3 jpm-12-00867-f003:**
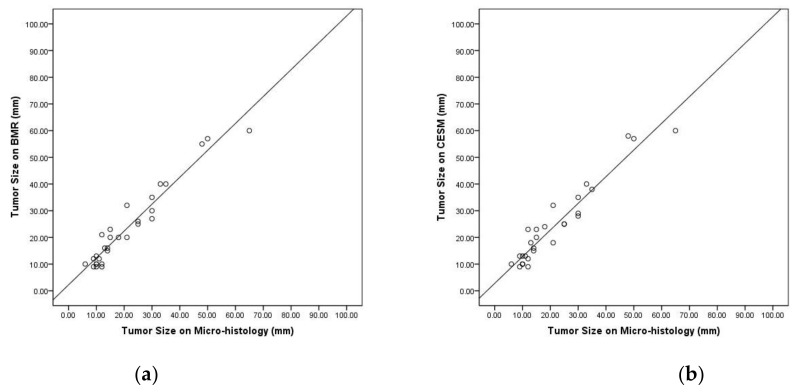
Pearson correlations of index cancers sizes of (**a**) breast magnetic resonance (r = 0.945, *p* < 0.001) and (**b**) contrast-enhanced spectral mammography (r = 0.937, *p* < 0.001) versus micro-histology.

**Figure 4 jpm-12-00867-f004:**
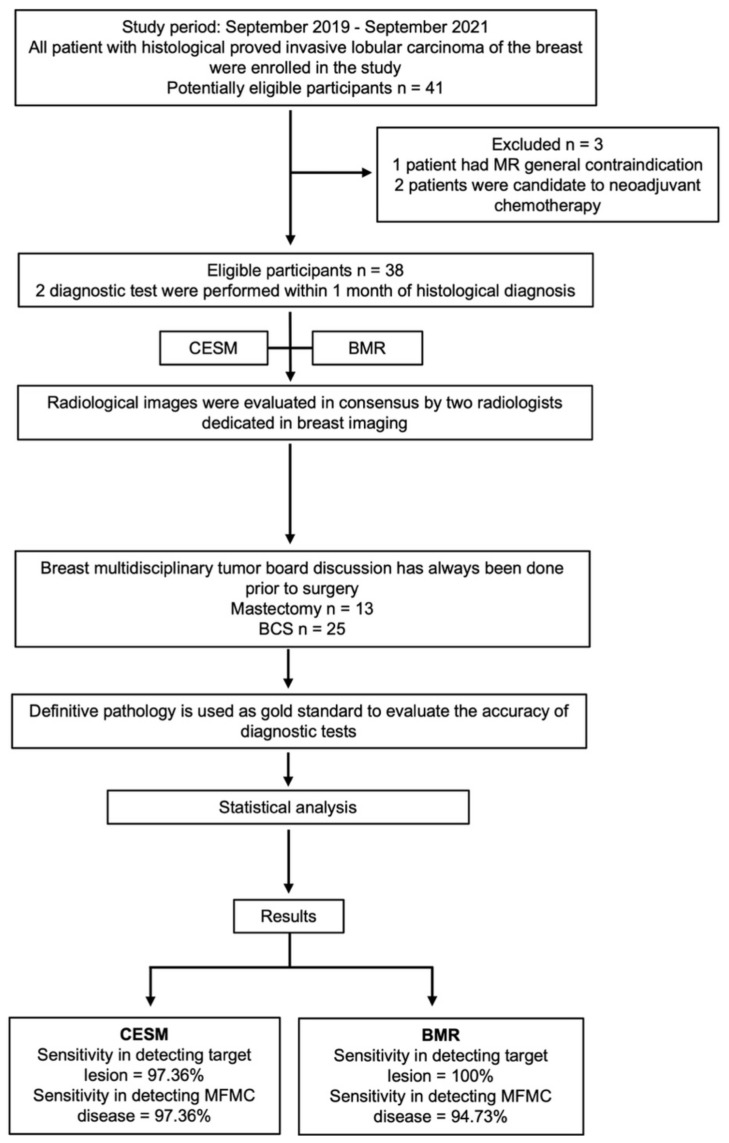
Strategy of the study, flowchart diagram. MR: magnetic resonance; CESM: contrast-enhanced spectral mammography; BMR: breast magnetic resonance; BCS: breast conserving surgery; MFMC: multifocal–multicentric.

**Figure 5 jpm-12-00867-f005:**
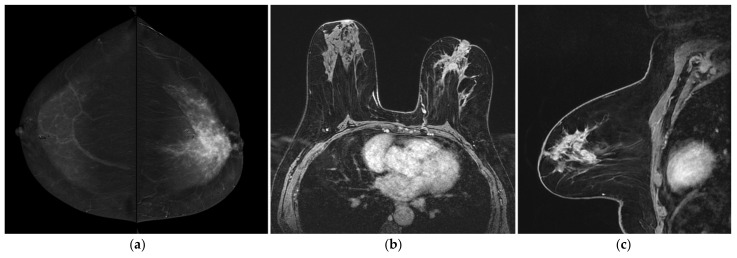
A 49-year-old women with large CLI of the left breast. Patient performs mammography because of a nipple retraction of the left breast. (**a**) CESM CC views show non-mass pathological enhancement on the left breast extending to the nipple. (**b**) Axial and (**c**) sagittal post-contrast T1-weighted fat saturation dynamic sequence (*Vibrant*) show large clustered ring regional non-mass enhancement consistent with pleomorphic CLI (T3 N2 M0, ER 90%, PgR 50%, Ki-67 35%, G3).

**Figure 6 jpm-12-00867-f006:**
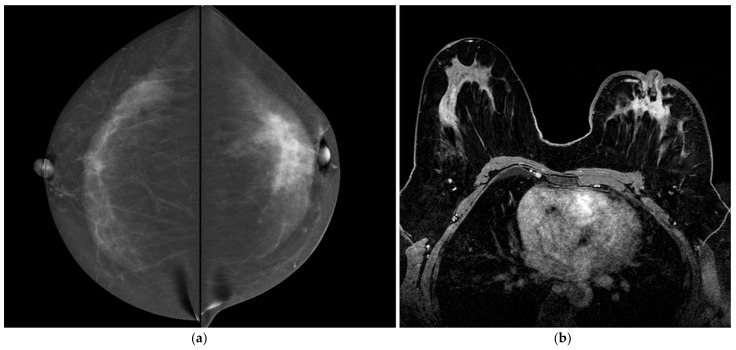
A 44-year-old women with large CLI of the left breast. Patient performs mammography because of a nipple retraction of the left breast. (**a**) CESM CC views show non-mass pathological enhancement on the left breast extending to the nipple. (**b**) Axial post-contrast T1-weighted fat saturation dynamic sequence (Vibrant) shows homogeneous non-mass enhancement consistent with classic CLI (T2, N1, M0, ER 90%, PgR 85%, Ki67 35%, G2). Pathological enhancement reaches the nipple. In right breast, BMR shows marked homogenous enhancement of the parenchyma of the external quadrants, that is not visible in CESM. Core needle biopsy was performed to confirm the absence of bilateral disease.

**Table 1 jpm-12-00867-t001:** Characteristics of patients.

Characteristics	Value
Number of included patients	38
Age, mean (range), years	59.7 (40–85)
Premenopausal, *n*. (%)	12 (31.6%)
Histopathological assessment, *n*. (%)	
Classic CLI	28 (73.7%)
Pleomorphic variant	10 (26.3%)
Unifocal disease	22 (57.9%)
Multifocal/Multicentric disease	16 (42.1%)
pTNM stage, *n*. (%)	
T1 N0	16 (42.1%)
T1 N1	3 (7.9%)
T1 N2	1 (2.6%)
T2 N0	11 (28.9%)
T2 N1	5 (13.2%)
T3 N2	2 (5.3%)
Molecular characteristics, *n*. (%)	
Luminal A	22 (57.9%)
Luminal B	14 (36.8%)
TN ^1^	2 (5.3%)

^1^ TN: Triple negative.

**Table 2 jpm-12-00867-t002:** CESM and BMR sensitivity.

Diagnostic Test	Sensitivity in Detecting Target Lesion			Sensitivity in Detecting MFMC Disease		
	TP	FN	TP/TP + FN	TP	FN	TP/TP + FN
**CESM**	37	1	37/38 = 97.36%	37	1	37/38 = 97.36%
**BMR**	38	0	38/38 = 100%	36	2	36/38 = 94.73%

MFMC = multifocal–multicentric; TP = true positive; FN = false negative. In CESM, there were 37 true positives and 1 false negative in the detection of target lesions for a sensitivity of 37/38 = 97.36%. In BMR, there were 38 true positives in the detection of target lesions for a sensitivity of 38/38 = 100%. In CESM, there were 37 true positives and 1 false negative in the detection of MFMC disease for a sensitivity of 37/38 = 97.36%. In BMR, there were 36 true positives and 2 false negatives in the detection of MFMC disease for a sensitivity of 36/38 = 94.73%.

**Table 3 jpm-12-00867-t003:** Means of the maximum diameter of target lesions assessed by BMR and CESM compared to histopathology.

	Mean (mm)	Range (mm)
BMR	25.94 (*sd* 16.99)	9–80
CESM	26.63 (*sd* 17.68)	9–85
Pathology	23.78 (*sd* 16.17)	6–75

## Data Availability

Data are contained within the article.

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
