# Peer review of "Diagnostic Challenge of Invasive Lobular Carcinoma of the Breast: What Is the News? Breast Magnetic Resonance Imaging and Emerging Role of Contrast-Enhanced Spectral Mammography"

_jpm, 2022, doi:10.3390/jpm12060867_

Round 1

Reviewer 1 Report

In their Systematic Research Article “ Diagnostic challenge in invasive lobular carcinoma of the breast: what are the news? Breast Magnetic Resonance Imaging and emerging role of Contrast-Enhanced Spectral Mammography" the Costantini et al. intended to incorporate the importance of CESM in assessing the ILC type of invasive BC.

This constitutes an interesting and broadly coherent body of work.

The article addresses a very relevant and of interest theme to the scientific community.

  1. The introduction portion is well defined and giving the rationale for conducting this study,
  2. The method and materials section is properly stated.
  3. The entire investigation was carried out with a well-designed study and yielded intriguing results.
  4. The authors have thoroughly highlighted the gathered data as well as the advantages of investigating the ability of CESM over BMR.

I appreciated the entire research experiment as well as the insightful points of view that the authors brought up in their investigation of CESM in the diagnosis of BC. 

Please consider revisiting the whole manuscript for minor spelling and grammar corrections.

Reviewer 2 Report

Melania Costantini et al. elaborated on the performance of contrast-enhanced spectral mammography (CESM) in breast invasive lobular carcinoma. They found that CESM displayed more accurate than breast MR in the diagnosis of invasive lobular cancers highlighting the encouraging usage of CESM. This study provides some insights on the applications of CESM in breast cancers but can be improved by the comments as below.

  1. The authors can provide a patient flow diagram to clarify the strategy.
  2. The authors should perform more statistical analysis on the performance characteristics between CESM and MR to reach out the conclusion (refer to PMID: 30240292).
  3. The authors should provide the pathological image information to verify the results.
